# Platinum-Induced Ototoxicity in Pediatric Cancer Patients: A Comprehensive Approach to Monitoring Strategies, Management Interventions, and Future Directions

**DOI:** 10.3390/children12070901

**Published:** 2025-07-08

**Authors:** Antonio Ruggiero, Alberto Romano, Palma Maurizi, Dario Talloa, Fernando Fuccillo, Stefano Mastrangelo, Giorgio Attinà

**Affiliations:** 1Pediatric Oncology Unit, Fondazione Policlinico Universitario Agostino Gemelli IRCCS, 00168 Rome, Italy; alberto.romano@guest.policlinicogemelli.it (A.R.); palma.maurizi@unicatt.it (P.M.); dario.talloa@guest.policlinicogemelli.it (D.T.); fernando.fuccillo@unicatt.it (F.F.); stefano.mastrangelo@unicatt.it (S.M.); giorgio.attina@policlinicogemelli.it (G.A.); 2Department of Woman and Child Health and Public Health, Università Cattolica del Sacro Cuore, 00168 Rome, Italy

**Keywords:** ototoxicity, platinum compounds, pediatric oncology, hearing loss, quality of life, monitoring guidelines, survivorship care

## Abstract

Platinum-induced ototoxicity constitutes a significant adverse effect in pediatric oncology, frequently resulting in permanent hearing impairment with profound implications for quality of life, language acquisition, and scholastic performance. This comprehensive review critically evaluates contemporary ototoxicity monitoring practices across various pediatric oncology settings, analyzes current guideline recommendations, and formulates strategies for implementing standardized surveillance protocols. Through examination of recent literature—encompassing retrospective cohort investigations, international consensus recommendations, and functional outcome assessments—we present an integrated analysis of challenges and opportunities in managing chemotherapy-associated hearing loss among childhood cancer survivors. Our findings demonstrate marked heterogeneity in monitoring methodologies, substantial implementation obstacles, and considerable impact on survivors’ functional status across multiple domains. Particularly concerning is the persistent absence of an evidence-based consensus regarding the appropriate duration of audiological surveillance for this vulnerable population. We propose a structured framework for comprehensive ototoxicity management emphasizing prompt detection, standardized assessment techniques, and integrated long-term follow-up care to minimize the developmental consequences of platinum-induced hearing impairment. This approach addresses critical gaps in current practice while acknowledging resource limitations across diverse healthcare environments.

## 1. Introduction

Childhood cancer survival rates have improved dramatically over recent decades, representing one of oncology’s most notable achievements. However, this expanding population of survivors confronts numerous treatment-associated late effects that substantially impact quality of life and functional status. Platinum-induced ototoxicity represents a particularly significant consequence for many pediatric solid tumor survivors. Platinum-based chemotherapeutic agents—primarily cisplatin and carboplatin—remain essential components in treatment regimens for numerous pediatric malignancies, including neuroblastoma, osteosarcoma, hepatoblastoma, and central nervous system tumors. While these agents have markedly enhanced survival outcomes, they are associated with substantial toxicities, with ototoxicity emerging as one of the most common and permanent adverse effects [1].

The documented incidence of platinum-induced hearing impairment in pediatric populations varies from 30% to 90%, contingent upon cumulative dosage, treatment protocol specifics, patient age, and assessment methodology [2,3]. This considerable variation reflects not only differences in treatment intensity but also inconsistencies in monitoring approaches and classification systems. Platinum compounds primarily exert ototoxic effects through damage to outer hair cells within the cochlea, initially affecting high-frequency regions with potential progression to involve lower frequencies essential for speech comprehension [4,5]. In contrast to adults, pediatric patients experiencing treatment-induced hearing loss face distinctive challenges during critical periods of language acquisition, cognitive development, and social skill formation. Consequently, even mild-to-moderate hearing deficits may have profound developmental implications that might not be immediately apparent but can manifest progressively throughout childhood and into adolescence.

Systematic monitoring for early detection has become recognized as a fundamental component of comprehensive cancer care for these patients. Timely identification of audiological changes can inform treatment modifications and facilitate prompt interventions that may minimize long-term consequences. However, translating this principle into consistent clinical practice has proven challenging across diverse healthcare environments globally [6,7,8]. A particularly concerning gap in current ototoxicity management is the absence of clear, evidence-based guidelines regarding the appropriate duration of audiological monitoring following completion of platinum-containing therapy. This uncertainty leaves many survivors vulnerable to undetected progressive hearing loss during crucial developmental periods.

This review examines contemporary practices in ototoxicity monitoring, evaluates hearing loss impact on survivors’ quality of life, analyzes existing guideline recommendations, and proposes strategies for implementing comprehensive monitoring and intervention programs. Through a synthesis of the recent literature, we aim to contribute to the ongoing refinement of survivorship care for children treated with platinum-based chemotherapy.

This narrative review synthesized current literature on platinum-induced ototoxicity in pediatric cancer survivors through systematic database searches. PubMed/MEDLINE, Embase, and Cochrane Library were searched from inception through December 2024 using MeSH terms and keywords combining pediatric oncology, platinum chemotherapy, and ototoxicity concepts. Studies were included if they involved pediatric patients (≤18 years) receiving cisplatin or carboplatin with audiological monitoring and outcomes reported. Adult-only populations, conference abstracts, and case reports with fewer than 10 patients were excluded. Given heterogeneity in study designs and outcome measures, narrative synthesis was employed to identify patterns in monitoring practices, guideline recommendations, and knowledge gaps.

## 2. Ototoxicity

### 2.1. Current State of Ototoxicity Monitoring

Despite widespread recognition of ototoxicity monitoring importance during and after platinum-based chemotherapy, the implementation of comprehensive protocols remains inconsistent across treatment centers. The retrospective analysis conducted by Diepstraten and colleagues at the Dutch National Cancer Center provides valuable insights into real-world monitoring practices [6]. Their evaluation of pediatric solid-tumor patients revealed that despite institutional protocols recommending audiological assessments before, during, and after platinum-based chemotherapy, actual implementation fell considerably short of recommendations. Baseline audiometry was conducted in approximately 70–85% of eligible patients, but regular monitoring during treatment occurred in only 50–65% of cases. Of particular concern was the observation that long-term follow-up assessments were completed in fewer than 50% of survivors, underscoring a significant gap in continuity of care [6].

This pattern is not isolated to a single institution but reflects broader challenges in implementing comprehensive monitoring protocols even in well-resourced healthcare environments. Multiple factors contribute to these implementation gaps, including resource limitations, competing clinical priorities, insufficient interdisciplinary collaboration, and knowledge deficits among healthcare providers regarding systematic monitoring importance [9]. Furthermore, the absence of universal consensus regarding monitoring frequency and duration contributes to substantial practice variability.

In addition, ototoxicity detection increasingly relies on specialized audiological techniques that extend beyond conventional pure-tone audiometry. Extended high-frequency (EHF) audiometry, testing frequencies from 8 to 20 kHz, demonstrates superior sensitivity for early cochlear damage detection, as platinum compounds preferentially affect basal cochlear regions responsible for high-frequency processing. While conventional audiometry may appear normal, EHF testing frequently reveals threshold elevation before speech-frequency involvement occurs, enabling earlier intervention.

Otoacoustic emissions testing, particularly distortion product otoacoustic emissions (DPOAEs), provides an objective assessment of outer hair cell function without requiring behavioral responses. This technique proves invaluable for monitoring young children or medically compromised patients where conventional audiometry may be unreliable. Serial DPOAE amplitude reductions often precede audiometric threshold changes, serving as sensitive indicators of emerging cochlear dysfunction [5,6,8].

Comprehensive assessment protocols should incorporate tympanometry to exclude middle ear pathology and acoustic reflex testing to evaluate retrocochlear function, particularly relevant given potential concurrent cranial irradiation effects. The integration of these objective measures with patient-reported outcomes provides a more complete assessment of functional hearing status than audiometric thresholds alone.

The International Late Effects of Childhood Cancer Guideline Harmonization Group, in collaboration with the PanCare Consortium, has made significant progress toward standardizing approaches to ototoxicity surveillance. Their comprehensive recommendations include audiological monitoring for all patients receiving platinum compounds, specifying baseline assessment prior to treatment initiation, follow-up assessments during treatment at defined intervals, and long-term monitoring extending beyond treatment completion [2]. However, these guidelines do not specify precisely for how long audiological follow-up should continue, leaving clinicians without clear direction regarding optimal monitoring duration.

This gap is particularly problematic given emerging evidence suggesting hearing loss may progress for years after treatment completion. Some survivors experience delayed-onset or progressive hearing deterioration that may remain undetected without continued monitoring well into adolescence and adulthood [1]. The absence of clear guidelines regarding follow-up duration means many survivors may be discharged from audiological surveillance prematurely, potentially missing opportunities for timely intervention during critical developmental periods.

### 2.2. Impact on Quality of Life and Development

The consequences of platinum-induced hearing loss extend considerably beyond audiological measurements, profoundly affecting multiple domains of survivors’ functioning. Rajput and colleagues conducted a comprehensive assessment of quality of life among pediatric cancer survivors with treatment-induced hearing loss, demonstrating impacts across educational, social, and psychological dimensions [4]. Their findings indicated that survivors with ototoxicity experienced significant difficulties in classroom environments, often struggling with instruction comprehension or full participation in group discussions. These challenges frequently necessitated educational accommodations and supportive services, potentially influencing academic achievement and educational trajectories [4].

Beyond academic settings, social interactions pose particular challenges for survivors with hearing impairment. Many reported difficulty following conversations in environments with background noise, leading to social withdrawal or reduced participation in age-appropriate activities. This social impact appeared especially pronounced during adolescence, when peer relationships assume heightened importance and social contexts become increasingly complex. The psychological burden of these challenges manifested in diminished self-confidence and increased self-consciousness regarding hearing difficulties or use of assistive devices [4,10,11].

Families of affected survivors also experienced significant stress related to communication difficulties, the management of hearing-assistive technologies, and the navigation of educational and healthcare systems to secure appropriate services. These findings underscore the importance of considering ototoxicity not merely as an audiological issue but as a complex challenge requiring comprehensive support across multiple functional domains. The magnitude of impact on quality of life correlated with hearing loss severity, with those experiencing moderate to severe impairment reporting more substantial challenges across multiple functional domains [4].

The developmental implications of these impacts are particularly concerning given the protracted nature of hearing loss in many survivors. Children who develop hearing deficits during critical periods of language acquisition may experience cascading effects on communication abilities, academic performance, and social development that may not become fully apparent until years after treatment completion. This developmental perspective further emphasizes the necessity for extended follow-up and comprehensive support services that evolve as survivors progress through different developmental stages.

### 2.3. Risk Factors and Prevention Strategies

Understanding factors predisposing patients to platinum-induced hearing loss is essential for developing targeted monitoring and prevention strategies. The Cochrane review by van As and colleagues identified several key risk factors associated with increased ototoxicity likelihood [1]. Higher cumulative platinum dose consistently emerged as a primary determinant of hearing loss risk, with studies demonstrating a dose-dependent relationship between platinum exposure and audiological outcomes [1]. Patient age at treatment represents another crucial factor, with younger children—particularly those under 5 years—demonstrating greater susceptibility to ototoxic effects. This age-related vulnerability likely reflects the developmental status of the auditory system and may have significant implications for long-term monitoring requirements in patients treated during early childhood [2].

Additional risk factors include concurrent cranial irradiation, which appears to potentiate platinum-induced cochlear damage through mechanisms potentially involving vascular alterations and oxidative stress. Simultaneous administration of other ototoxic medications, including certain aminoglycoside antibiotics and loop diuretics, may increase risk through additive or synergistic effects [1,12,13]. Emerging research suggests genetic susceptibility factors, including polymorphisms in genes involved in platinum metabolism and detoxification pathways, may contribute to individual variation in ototoxicity risk, though these findings require further validation before clinical application [2].

Prevention strategies discussed in the literature include several approaches, though efficacy evidence varies substantially. Dose modifications based on early detection of hearing changes represent a common strategy, though this approach necessitates careful balance between ototoxicity risk and the potential compromise of antineoplastic efficacy. Some protocols incorporate extended infusion durations to reduce peak platinum concentrations, though evidence supporting this approach remains limited. The administration of otoprotective agents has been investigated extensively, with compounds such as sodium thiosulfate demonstrating promise in specific clinical contexts. However, concerns regarding potential interference with antitumor activity have limited widespread implementation of these agents [1,14,15].

The identification of these risk factors and potential preventive strategies underscores the importance of comprehensive risk assessment and individualized monitoring approaches. Patients with multiple risk factors may benefit from more intensive surveillance protocols, while emerging genetic information may eventually facilitate more personalized risk stratification. However, the variable implementation of even basic monitoring protocols suggests that substantial barriers exist to translating this knowledge into consistent clinical practice.

### 2.4. Guidelines and Recommendations: Addressing the Follow-Up Duration Gap

Significant progress has been achieved in developing evidence-based recommendations for ototoxicity monitoring, primarily through the work of the International Late Effects of Childhood Cancer Guideline Harmonization Group. Their comprehensive guidelines, published by Clemens and colleagues, provide detailed recommendations regarding monitoring candidates, assessment methodologies, and evaluation timing [2]. These guidelines recommend audiological assessment for all patients receiving platinum-based chemotherapy, with baseline evaluation prior to treatment initiation and follow-up assessments during and after treatment completion.

The guidelines emphasize the importance of utilizing age-appropriate testing methods based on developmental stage and consistent classification using validated grading systems such as SIOP Boston, Chang, or CTCAE criteria. This standardized approach aims to enhance comparability across studies and facilitate clearer communication among clinicians regarding hearing loss severity and progression [2].

However, a critical limitation of current guidelines is the absence of specific recommendations regarding the appropriate duration of audiological monitoring following treatment completion. While the guidelines suggest monitoring should extend into survivorship, they do not specify concrete timeframes or criteria for discontinuing surveillance. This gap is particularly problematic given evidence suggesting that hearing loss may progress over time, with some patients experiencing deterioration years after treatment completion [1,16,17,18,19].

The absence of clear guidance regarding follow-up duration has several important implications. First, it contributes to practice variability, with some centers continuing monitoring indefinitely while others discontinue surveillance after a defined period, regardless of individual risk profile. Second, it complicates transition planning for survivors transferring from pediatric to adult healthcare settings, potentially resulting in discontinuity of audiological care during this vulnerable period. Finally, it creates challenges for resource allocation and service planning, as centers attempt to determine appropriate staffing and infrastructure for long-term follow-up programs.

Fernandez and colleagues have recently addressed this gap through their proposed framework for global implementation of ototoxicity management [9]. Their approach emphasizes the necessity for lifelong monitoring for certain high-risk groups, suggesting surveillance should continue indefinitely for survivors who received high cumulative doses, underwent treatment at very young ages, or demonstrate progressive hearing changes during initial follow-up. For lower-risk groups, they propose risk-stratified approaches that consider individual factors in determining appropriate monitoring duration [9].

This approach represents an important advancement toward addressing the follow-up duration question, though further research and consensus-building efforts are needed to develop more specific, evidence-based guidelines in this area. Longitudinal studies tracking hearing outcomes over extended periods will be essential to inform these recommendations and ensure monitoring resources are allocated efficiently to those at greatest risk for progressive or delayed-onset hearing loss.

## 3. Toward a Comprehensive Approach: Integrating Monitoring, Intervention, and Support

Addressing platinum-induced ototoxicity challenges requires progression beyond fragmented approaches to develop integrated systems spanning from early detection to long-term support. From the literature reviewed, several key elements emerge as essential components of comprehensive ototoxicity management programs (Table 1).

Standardized assessment protocols constitute the foundation for ensuring consistent detection and classification of hearing changes. These protocols should incorporate developmentally appropriate testing methods, validated grading systems, and clear documentation standards accessible to multidisciplinary teams [2]. Implementation should be supported by healthcare provider education initiatives, integration into electronic health records, and regular quality audits to identify and address compliance barriers.

Risk-stratified monitoring approaches offer opportunities to optimize resource allocation while ensuring appropriate surveillance for all patients. Patients with high-risk characteristics—such as young treatment age, high cumulative platinum doses, or concurrent risk factors—may benefit from more intensive monitoring schedules and extended follow-up [9]. As genetic susceptibility testing becomes more refined, these approaches may eventually incorporate genomic information to further individualize monitoring plans.

Multidisciplinary management is essential to address the multifaceted impacts of hearing loss. Collaboration among oncology, audiology, speech–language pathology, psychology, and educational specialists can facilitate comprehensive assessment and intervention planning that addresses not only audiological outcomes but also functional impacts on communication, learning, and psychosocial well-being [4,20,21,22,23,24]. This multidisciplinary approach is particularly crucial during key developmental transitions, when changing demands may reveal previously unrecognized functional limitations.

The management of platinum-induced hearing loss requires consideration of diverse technological interventions tailored to individual severity and functional needs. Modern digital hearing aids offer sophisticated signal processing capabilities particularly beneficial for the high-frequency hearing loss patterns characteristic of ototoxicity. Features including directional microphones, noise reduction algorithms, and frequency-lowering technology can significantly improve speech understanding in challenging acoustic environments that disproportionately affect survivors with hearing impairment [25].

Educational environments present unique challenges requiring specialized amplification approaches. Frequency modulation and digital modulation systems deliver teacher voice directly to hearing aids or dedicated receivers, overcoming classroom acoustics limitations and background noise interference. These systems prove essential for maintaining academic participation, particularly during critical developmental periods when educational demands increase [26].

For survivors developing severe-to-profound hearing loss, cochlear implantation represents a viable intervention option. While temporal bone anatomy may be altered by previous treatment, emerging evidence suggests that outcomes in pediatric cancer survivors can parallel those achieved in children with congenital hearing loss. Candidacy evaluation must consider concurrent medical conditions, developmental stage, and rehabilitation potential within the context of ongoing survivorship care [27,28].

Long-term survivorship care must incorporate ongoing attention to hearing health throughout the lifespan. This requires not only extended audiological monitoring but also systematic education for survivors and families regarding hearing preservation strategies, communication techniques, and available interventions [9]. Transition planning for survivors transferring from pediatric to adult healthcare settings should specifically address audiological care continuity, with clear communication between providers regarding monitoring history and ongoing requirements.

Quality improvement initiatives are needed to continuously refine monitoring protocols and intervention approaches based on emerging evidence and implementation experience. Regular audits of monitoring compliance and outcomes can identify system-level barriers and inform targeted improvement strategies. The incorporation of patient-reported outcome measures can provide valuable insights regarding the functional impact of hearing changes and intervention effectiveness from the survivor perspective [4,29,30].

The implementation of these elements will require coordinated efforts across multiple levels, from individual providers to healthcare systems to policy-making bodies. Resource constraints represent a significant challenge, particularly in settings with limited access to specialized audiological services.

Innovative approaches such as teleaudiology, mobile screening programs, and task-shifting strategies may help address these constraints while maintaining quality standards [9]. Advocacy efforts highlighting the developmental and quality of life impacts of untreated hearing loss may help secure necessary resources for comprehensive programs.

## 4. Conclusions and Future Perspectives

Platinum-induced ototoxicity represents a significant challenge in pediatric oncology, with far-reaching implications for survivors’ development and quality of life. While substantial progress has been achieved in understanding risk factors, developing monitoring guidelines, and recognizing quality of life impacts, significant gaps remain in translating this knowledge into consistent clinical practice. The absence of clear guidelines regarding follow-up duration represents a particularly concerning deficiency that leaves many survivors vulnerable to undetected progressive hearing loss during critical developmental periods.

Several priorities emerge for future research and clinical practice development. Longitudinal studies tracking hearing outcomes over extended timeframes are urgently needed to inform evidence-based recommendations regarding optimal monitoring duration for different risk groups. The implementation of scientific approaches can help identify and address barriers to guideline adherence, particularly in resource-constrained environments. The investigation of novel otoprotective approaches that preserve antitumor efficacy remains an important goal for reducing ototoxicity incidence. The refinement of risk prediction models, potentially incorporating genetic markers, may eventually enable more personalized approaches to monitoring and prevention.

The establishment of international registries that record audiological outcomes across diverse treatment protocols and healthcare settings could accelerate knowledge generation and facilitate continuous quality improvement. The development of interventional studies addressing functional impacts of hearing loss could help identify optimal approaches for supporting affected survivors across different developmental stages. Finally, research examining the cost-effectiveness of various monitoring and intervention strategies could inform resource allocation decisions and policy development.

Through collaborative efforts spanning clinical care, research, and education, the oncology community can work toward minimizing hearing loss impact on the expanding population of childhood cancer survivors. While challenges remain substantial, the potential benefits—improved developmental trajectories, enhanced educational outcomes, and superior quality of life—make this an essential component of comprehensive cancer survivorship care.

## Figures and Tables

**Table 1 children-12-00901-t001:** Risk factors, assessment, and management strategies for platinum-induced ototoxicity in pediatric cancer survivors.

Category	Components	Clinical Considerations
Risk Factors	*Patient-related*	
	Age < 5 years at treatment	Higher vulnerability due to auditory system development; requires more intensive monitoring
	Pre-existing hearing impairment	Potential for compounded hearing deficits; baseline assessment critical
	Genetic polymorphisms (TPMT, COMT, GST)	Emerging evidence for genetic susceptibility; potential for future risk stratification
	*Treatment-related*	
	High cumulative cisplatin dose (>360 mg/m^2^)	Strong dose–response relationship; consider dose modifications when possible
	Bolus administration (vs. extended infusion)	Higher peak plasma concentrations associated with increased risk
	Concurrent cranial radiotherapy	Synergistic ototoxic effects; requires enhanced monitoring
	Concomitant ototoxic medications	Aminoglycosides, loop diuretics may potentiate effects; avoid when possible
Assessment Protocols	*Timing*	
	Baseline (pre-treatment)	Essential for detecting pre-existing hearing loss; affects treatment planning
	During treatment (after each cycle)	Enables early detection and potential treatment modifications
	End of treatment	Documents acute ototoxicity; establishes baseline for long-term follow-up
	Long-term follow-up	Critical for detecting delayed/progressive loss; optimal duration undefined
	*Methodology*	
	Age-appropriate audiological testing	Selection based on developmental status; ensures valid assessment
	Extended high-frequency audiometry	Enhances sensitivity for early detection (when developmentally appropriate)
	Standardized grading systems	SIOP Boston, Chang, or CTCAE criteria; enables consistent classification
	Patient/parent-reported outcome measures	Captures functional impact; supplements audiometric data
Management Approaches	*Prevention*	
	Otoprotective agents (e.g., sodium thiosulfate)	Promising results; concerns regarding interference with antitumor efficacy
	Treatment protocol modifications	Dose adjustments based on early audiological changes
	Extended infusion regimens	May reduce peak concentrations; evidence limited
	*Intervention*	
	Hearing assistive technology	Hearing aids, FM systems, cochlear implants based on loss severity
	Educational accommodations	Preferential seating, classroom acoustics optimization, additional support
	Communication strategy training	Benefits both survivors and families; enhances functional outcomes
	Psychosocial support	Addresses self-esteem and social challenges related to hearing loss
Monitoring Program Components	*Structural elements*	
	Multidisciplinary team approach	Oncology, audiology, SLP, psychology, education specialists
	Risk-stratified protocols	Monitoring intensity/duration based on individual risk profile
	Electronic health record integration	Enhances compliance, facilitates communication among providers
	Transition planning	Ensures continuity when transferring to adult healthcare
	Quality improvement mechanisms	Regular audits, outcome tracking, protocol refinement

Legend: TPMT = thiopurine methyltransferase; COMT = catechol-O-methyltransferase; GST = glutathione S-transferase; SIOP = International Society of Pediatric Oncology; CTCAE = Common Terminology Criteria for Adverse Events; FM = frequency modulation; SLP = speech–language pathology.

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
