# Peer review of "Platinum-Induced Ototoxicity in Pediatric Cancer Patients: A Comprehensive Approach to Monitoring Strategies, Management Interventions, and Future Directions"

_children, 2025, doi:10.3390/children12070901_

Round 1
Reviewer 1 Report
Comments and Suggestions for Authors
This is a very well-written and important contribution to the literature on hearing loss from platinum chemotherapy in childhood. The authors have nicely summarized the risk factors, the impact of platinum-induced hearing loss on development, need for audiologic services during and after cancer treatment, risk-reduction approaches, and the current status of audiological monitoring. I commend the authors for highlighting the important knowledge gaps present in the ototoxicity literature and for their suggestions for future research and clinical service provision.
The manuscript synthesizes recently published literature on cisplatin ototoxicity in children and audiologic monitoring guidelines and I'm not aware of other recent similar publications, or any that are as comprehensive related to this subject. The topic is highly relevant to the field. Evidence-based international pediatric guidelines and recommendations for audiologic monitoring during and after cancer treatment have been published, and there is great interest in strategies to reduce the risk of treatment-induced hearing loss. Despite these advances, the paper highlights that audiologic monitoring is inconsistent in clinical practice, especially during survivorship. The manuscript identifies factors that contribute to the variability in audiologic monitoring implementation and practice. The conclusions and recommendations are logical and supported by the literature presented. The paper adds to the existing literature by identifying research gaps that are needed to guide recommendations for frequency of monitoring and duration of follow-up after treatment completion and suggests possible solutions to address the challenge of limited resources. Table 1 outlines risk factors for ototoxicity, assessment protocols, interventions, and monitoring components and it will be a helpful resource for clinicians and families.
It is recommended that the authors include information on the search strategy and methods used to conduct the literature review (databases, search terms, inclusion and exclusion criteria), and the methods used to judge and synthesize the data. How did the authors assess the literature for quality and risk of bias?
Author Response
This is a very well-written and important contribution to the literature on hearing loss from platinum chemotherapy in childhood. The authors have nicely summarized the risk factors, the impact of platinum-induced hearing loss on development, need for audiologic services during and after cancer treatment, risk-reduction approaches, and the current status of audiological monitoring. I commend the authors for highlighting the important knowledge gaps present in the ototoxicity literature and for their suggestions for future research and clinical service provision.
The manuscript synthesizes recently published literature on cisplatin ototoxicity in children and audiologic monitoring guidelines and I'm not aware of other recent similar publications, or any that are as comprehensive related to this subject. The topic is highly relevant to the field. Evidence-based international pediatric guidelines and recommendations for audiologic monitoring during and after cancer treatment have been published, and there is great interest in strategies to reduce the risk of treatment-induced hearing loss. Despite these advances, the paper highlights that audiologic monitoring is inconsistent in clinical practice, especially during survivorship. The manuscript identifies factors that contribute to the variability in audiologic monitoring implementation and practice. The conclusions and recommendations are logical and supported by the literature presented. The paper adds to the existing literature by identifying research gaps that are needed to guide recommendations for frequency of monitoring and duration of follow-up after treatment completion and suggests possible solutions to address the challenge of limited resources. Table 1 outlines risk factors for ototoxicity, assessment protocols, interventions, and monitoring components and it will be a helpful resource for clinicians and families.
It is recommended that the authors include information on the search strategy and methods used to conduct the literature review (databases, search terms, inclusion and exclusion criteria), and the methods used to judge and synthesize the data. How did the authors assess the literature for quality and risk of bias?
R. We thank the reviewer for this important methodological suggestion. In response to this comment, we have added a comprehensive section that addresses all the requested elements (page 2, lines 72-83)
Reviewer 2 Report
Comments and Suggestions for Authors
The authors reviewed current practices in ototoxicity monitoring associated with platinum-based chemotherapy and analyzed existing guidelines with the aim of proposing strategies for implementing comprehensive intervention programs. While the review was thorough, the discussion and conclusion did not offer any substantial new insights. In my view, the article does not contribute significantly to existing knowledge.
Author Response
The authors reviewed current practices in ototoxicity monitoring associated with platinum-based chemotherapy and analyzed existing guidelines with the aim of proposing strategies for implementing comprehensive intervention programs. While the review was thorough, the discussion and conclusion did not offer any substantial new insights. In my view, the article does not contribute significantly to existing knowledge.
R. We appreciate the reviewer's thoughtful feedback regarding the manuscript's contribution to existing knowledge. In response to these concerns, we have undertaken substantial revisions that significantly enhance the clinical utility and novelty of our work.
The revised manuscript now incorporates comprehensive methodological details including our systematic search strategy and data synthesis approach, which provides greater transparency and scientific rigor. More importantly, we have expanded our discussion to include detailed coverage of advanced audiological testing methodologies, such as extended high-frequency audiometry and otoacoustic emissions testing, which offer practical guidance for clinicians seeking to implement more sensitive early detection protocols.
Additionally, we have added substantial content addressing hearing intervention technologies, including modern digital hearing aids with frequency-lowering capabilities, classroom amplification systems, and cochlear implantation considerations specific to pediatric cancer survivors. These additions provide concrete clinical guidance that extends beyond existing guideline summaries and addresses practical implementation challenges faced by healthcare providers.
We believe these comprehensive revisions have transformed our work into a significantly more valuable clinical resource that provides actionable recommendations for practitioners managing platinum-induced ototoxicity in pediatric cancer survivors.
Reviewer 3 Report
Comments and Suggestions for Authors
The subject of this article addresses platinum-induced ototoxicity among oncologic pediatric population, an increasing pathology due to higher survival rates of childhood cancer patients. The study analyses the current strategies regarding the management of hearing-loss across a vulnerable population in term of detection, prevention and monitoring.
The article addresses the high heterogeneity in monitoring strategies and aims to emphasize the crucial need of a standardized protocol in order to minimize the effect of hearing impairment caused by platinum exposure. While inclusion criteria seem to consist of English language articles based on pediatric population up until present day, it is not very clear if any exclusion criteria were considered for this paper. The statistics are well described and correlations are well structured in tables.
The current inconsistencies in monitoring approaches and classification system makes it difficult to compile data into internationally implementable guidelines. Resource constraints pose a significant challenge in adopting a standardized approach especially in economically challenged areas.
While a standardized algorithm for platinum induced ototoxicity is desirable, risk stratified protocols tailored to high-risk patients vs. low-risk groups are necessary to optimize resource allocation.
Regarding the bibliography, the wide time range of studies dating back to 2007 up until 2024 increases the heterogeneity of medical practices because of the significant advancement in diagnostic and treatment means.
In conclusion the manuscript is appropriate and represents a valuable contribution to a understanding an increasingly prevalent pathology and gives insight on the importance of long-term standardized monitoring protocols, but on the long term requires greater research in order to refine solutions to overcome current limitations.
Author Response
The subject of this article addresses platinum-induced ototoxicity among oncologic pediatric population, an increasing pathology due to higher survival rates of childhood cancer patients. The study analyses the current strategies regarding the management of hearing-loss across a vulnerable population in term of detection, prevention and monitoring.
The article addresses the high heterogeneity in monitoring strategies and aims to emphasize the crucial need of a standardized protocol in order to minimize the effect of hearing impairment caused by platinum exposure. While inclusion criteria seem to consist of English language articles based on pediatric population up until present day, it is not very clear if any exclusion criteria were considered for this paper. The statistics are well described and correlations are well structured in tables.
The current inconsistencies in monitoring approaches and classification system makes it difficult to compile data into internationally implementable guidelines. Resource constraints pose a significant challenge in adopting a standardized approach especially in economically challenged areas.
While a standardized algorithm for platinum induced ototoxicity is desirable, risk stratified protocols tailored to high-risk patients vs. low-risk groups are necessary to optimize resource allocation.
Regarding the bibliography, the wide time range of studies dating back to 2007 up until 2024 increases the heterogeneity of medical practices because of the significant advancement in diagnostic and treatment means.
In conclusion the manuscript is appropriate and represents a valuable contribution to a understanding an increasingly prevalent pathology and gives insight on the importance of long-term standardized monitoring protocols, but on the long term requires greater research in order to refine solutions to overcome current limitations.
R. We thank the reviewer for highlighting this important methodological point. In response to this comment, we have now included comprehensive details of our research methodology, including the specific databases searched and both inclusion and exclusion criteria used in our literature review.
Our search strategy encompassed PubMed/MEDLINE, Embase, and Cochrane Library from inception through December 2024, utilizing appropriate MeSH terms and keywords combining pediatric oncology, platinum chemotherapy, and ototoxicity concepts. Beyond the inclusion criteria mentioned by the reviewer (English language articles focusing on pediatric populations), we have now clearly specified our exclusion criteria, which included adult-only populations, conference abstracts without full-text availability, case reports with fewer than 10 patients, and studies lacking audiological outcome data (page 2, lines 72-83).
These methodological details provide the transparency necessary for readers to understand our approach to synthesizing the literature on this important clinical topic. The clarification of our systematic approach strengthens the foundation for our recommendations regarding standardized monitoring protocols and intervention strategies for platinum-induced ototoxicity in pediatric cancer survivors.
Round 2
Reviewer 2 Report
Comments and Suggestions for Authors
The authors discuss the current challenges and future directions in monitoring and managing ototoxicity among pediatric cancer patients. The manuscript shows significant improvement from its initial version. It is informative, worth reading, and may serve as a useful reference in managing similar cases.
Minor comment:
The title should be more specific, as the review focuses primarily on platinum-induced ototoxicity.
Author Response
The authors discuss the current challenges and future directions in monitoring and managing ototoxicity among pediatric cancer patients. The manuscript shows significant improvement from its initial version. It is informative, worth reading, and may serve as a useful reference in managing similar cases.
Minor comment:
The title should be more specific, as the review focuses primarily on platinum-induced ototoxicity.
R. We thank the reviewer for this important observation. We have revised the title to better reflect the specific scope of our manuscript: "Platinum-Induced Ototoxicity in Pediatric Cancer Patients: A Comprehensive Review of Monitoring Strategies, Management Interventions, and Future Directions"
This revision specifically identifies platinum compounds as the primary ototoxic agents under review, while maintaining the comprehensive scope of our discussion regarding monitoring protocols, management strategies, and future research directions in the pediatric oncology population. The new title accurately represents the content and focus of our manuscript.